# Recent Applications of Contact Lenses for Bacterial Corneal Keratitis Therapeutics: A Review

**DOI:** 10.3390/pharmaceutics14122635

**Published:** 2022-11-28

**Authors:** Linyan Nie, Yuanfeng Li, Yong Liu, Linqi Shi, Huiyun Chen

**Affiliations:** 1Department of Ophthalmology, The People’s Hospital of Yuhuan, Yuhuan 317600, China; 2Oujiang Laboratory (Zhejiang Lab for Regenerative Medicine, Vision and Brain Health), Wenzhou Institute, University of Chinese Academy of Sciences, Wenzhou 325001, China

**Keywords:** bacterial keratitis, bacterial resistance, functional contact lens, bacterial killing

## Abstract

Corneal keratitis is a common but severe infectious disease; without immediate and efficient treatment, it can lead to vision loss within a few days. With the development of antibiotic resistance, novel approaches have been developed to combat corneal keratitis. Contact lenses were initially developed to correct vision. Although silicon hydrogel-based contact lenses protect the cornea from hypoxic stress from overnight wear, wearing contact lenses was reported as an essential cause of corneal keratitis. With the development of technology, contact lenses are integrated with advanced functions, and functionalized contact lenses are used for killing bacteria and preventing infectious corneal keratitis. In this review, we aim to examine the current applications of contact lenses for anti-corneal keratitis.

## 1. Introduction

Infectious keratitis remains a significant cause of visual loss or blindness globally. It may be caused by infectious bacteria, fungi, viruses, parasites, protozoa, and bacterial contaminants [1]. Bacterial keratitis is an acute or chronic transient corneal lesion requiring immediate, efficient treatment and appropriate follow-up. Bacterial keratitis is widely accepted as an infection caused by bacterial pathogens, especially multi-drug-resistant bacteria, due to antibiotic abuse. As there are no simple, direct, and standard diagnosis approaches available for infectious keratitis, traditional treatments with antibiotics often cause drug resistance. Therefore, novel methods are in high demand to combat bacterial keratitis. Contact lenses (CLs) are primarily designed for vision correction. However, there are increasing ocular complications among contact wearers; one is associated with bacterial keratitis resulting from the overnight wearing of CLs. With the improvement of nanotechnologies, various materials are loaded in CL hydrogels to meet different requirements. Nowadays, CLs are recruited in various, more advanced medical applications, including drug delivery [2], sensing biomarkers for cancer or diabetes [3], intraocular pressure measurement [4], dry eye disease [5], color blindness management [6], etc. There are CLs with antibacterial abilities that are developed in order to prevent infectious keratitis in the first instance for CL wearers. There are several strategies available for the fabrication of CLs with bacterial inhibition functions, such as loading nanomaterials (e.g., metallic nanoparticles including gold, silver, etc.), adding antibacterial peptides, or integrating agents that can release reactive oxygen species (ROS) with a controllable dose that is sufficient to kill bacteria without causing adverse effects. These methods are able to kill bacteria or prevent bacterial adherence, thus avoiding bacterial keratitis. In addition, they do not cause drug resistance and they prevent antibiotic abuse. This review summarizes the approaches to the current diagnosis and traditional treatment of bacterial keratitis and focuses on recent applications using functionalized contact lens-based antibacterial applications for bacterial keratitis therapy.

## 2. Diagnosis of Bacterial Keratitis

Successful bacterial management depends on accurate, timely diagnosis and proficient interventions [7]. Unlike many other diseases, there is no reliable reference standard for diagnosing corneal keratitis. Thus, it is difficult to verify the tests on suspected bacterial keratitis. Several methods are available now for bacterial keratitis diagnosis, including colony-based microbiological investigations, corneal imaging, and molecular detection. Here, we provide an overview of the fundamental principles, diagnosis performance, advantages, and drawbacks of the state of the art of current diagnosis methods.

### 2.1. Traditional Colony Culture Investigation

To detect suspected bacterial pathogens, a ‘corneal scrape’ is performed to collect samples that contain causative organisms [8]. The sample is directly inoculated into the culture media and immediately transferred to the microbiology lab for Gram staining. With staining by a water-soluble dye named crystal violet, scientists can differentiate bacterial stains into two large groups based on their different cell wall constituents: Gram-positive and Gram-negative bacteria. Gram-positive and Gram-negative bacteria will be rendered pink and blue, respectively. The Gram stains permit the initial treatment with antibiotics.

### 2.2. Corneal Imaging

When the colony cultures are repeatedly negative, it is necessary to perform corneal imaging or a so-called corneal biopsy [9,10]. The corneal biopsy is usually performed under a slit lamp in the operation room. The obtained corneal tissue is then sent for culture and histopathological analysis. Depending on the clinical features and the sample amount, the corneal samples are processed using electron and light microscopy, immunofluorescence, histochemistry, or histopathologic analysis [11,12]. 

### 2.3. Molecular Detection

The standard laboratory culture techniques fail to identify some slow-growing bacteria, which is time-consuming. With the collected samples from the ‘corneal scrape’, one can also run molecular tests by polymerase chain reaction (PCR) or next-generation sequencing (NGS) to identify or characterize the microorganisms. To perform PCR tests, the samples must be collected by sterile swabs. PCR is a highly sensitive technique that allows the rapid amplification of tiny DNA samples but is easily affected by the cotton swab [13]. NSG allows the massive sequencing of nucleic acid by high-throughput sequencing technologies. The use of NGS offers high accuracy when directly diagnosing clinical samples. However, test validation, high costs, and reproducibility are routine drawbacks and limit their use in clinical trials [14]. 

### 2.4. Signs and Symptoms

The common symptoms of bacterial keratitis include ocular pain, photophobia, red eye, tearing, conjunctival mucopurulent discharge, and a variable extent of vision loss [14]. Bacterial keratitis often leads to robust inflammation responses due to the ocular immune response to bacterial pathogens and their products (e.g., toxins), trauma, and allergies. 

### 2.5. Traditional Treatments

Prior to contact lens application, wearers should be warned or appropriately educated. For the initial treatment, broad-spectrum antibiotics are commonly used as the first line of therapy to treat bacterial keratitis. However, only limited choices of licensed antibiotics are available for topical use [15]. The clearance from the tear film or frequently applied eye drops or ointments makes the administrated dose of antibiotics irrelevant to the active biological dose of antibiotics. Moreover, some molecules (e.g., glycopeptides) are not even able to penetrate the cornea and kill bacteria. 

## 3. Contact Lenses for Corneal Keratitis

The clinical needs of bacterial keratitis remain unmet, to avoid antibiotic abuse. Hydrogel-based CLs could be a promising option for achieving noninvasive, low-cost, and easy-to-use therapeutic approaches to corneal keratitis. Nowadays, CLs are applied beyond their primary role in vision correction. They also play an essential role in medical devices and are recruited in advanced medical applications, as mentioned above. There are currently several different solutions using CLs to combat bacterial keratitis, including the use of contact lenses to deliver drugs and obtain sustained drug release compared to eye drops; polymerizing nanoparticles (e.g., silver, gold), sugar (e.g., chitosan), or peptides into the hydrogel substrates; and creating CL gels endowed with the controllable release of nitric oxide (NO), which kills bacteria. 

### 3.1. CL Hydrogel Types

Since the invention of CLs, they have confronted and overcome several challenges, including hypoxia, low moisture, etc. Therefore, these are critical when selecting the substrate materials for CLs. The materials must permit the CLs to possess features such as oxygen permeability, moisturization, outstanding visible light transmittance, and stability to maintain the lens structure, and they must ensure comfort for the wearer [16]. Hydrogels including poly (methyl methacrylate) (PMMA), poly (ethylene terephthalate) (PET), poly (2-hydroxyethyl methacrylate) (pHEMA), polydimethylsiloxane (PDMS), silicone, and 2-methacryloyloxyethyl phosphorylcholine (MPC) are used to polymerize contact lenses. Figure 1 shows that CLs are classified into hard and soft CLs. Hard CLs are rigid with high gas permeability; however, they have low flexibility on the eyes. Soft CLs are highly flexible in the eyes, with limited oxygen permeability. PMMA and PET are often used for polymerizing hard CLs. Attributed to its several advantages, PMMA is an ideal material for polymerizing hard CLs, e.g., it has outstanding optical transparency, ease of fabrication, facility of sterilization, and a low cost [17]. However, PMMA contact lenses only allow limited oxygen permeability. PET is another material used for hard CLs; it is also commonly used for plastic bottles. It possesses good chemical and heat resistance, and therefore it is easy to shape into various complex structures [17]. PDMS, MPC, HEMA, and silicone hydrogels are commonly used for polymerizing soft CLs. PDMS is a polymer that is transparent, elastic, air-permeable [18], and biocompatible. The most important feature of PDMS is that it is biocompatible, which is important for biomedical applications. Due to the biological inertness of the phosphorylcholine group, MPC is resistant to protein adsorption, cell adhesion, and blood coagulation [19], making it a popular material for contact lenses. In addition, it also has been used in many medical devices, such as artificial hip joints [20] and rapid disease diagnosis systems [20,21]. HEMA is a high-performance material used in biomedical applications; it has been used to develop artificial corneal [22] and bone tissue [23]. Because of its great optical transparency, gas permeability, and biocompatibility, it is an ideal material for soft CLs. When using HEMA hydrogels to fabricate CLs, the CLs can obtain high water content (38%), and they maintain good wettability [17]. In addition, HEMA hydrogels can be copolymerized with many other monomers to obtain different CLs with unique functions [17]. For instance, HEMA can be copolymerized with ethylene glycol dimethacrylate to achieve the kinetic release of triamcinolone acetonide [24]. Silicone hydrogels are widely used in ophthalmic applications, thanks to their distinctive characteristics of biocompatibility, optical transparency, and oxidative and thermal stability [25]. In addition, as silicone hydrogels contain both water and siloxane channels, the oxygen permeability of CLs is permitted [26]. 

Table 1 summarizes the advantages and disadvantages of these materials. Among these materials, silicone and HEMA hydrogels are ideal for multifunctional contact lenses, and there are several factors that need to be considered when selecting CL materials, including oxygen permeability, wettability, mechanical properties, and costs. For example, D. H. Keum et al., using silicone hydrogels as CL materials, developed remotely controllable smart CLs that can monitor glucose levels as well as treat diabetic retinopathy with sustained drug release [27]. S. Li et al. used HEMA hydrogels as CL materials and embedded them with antibody-conjugated signaling microchambers that could detect tear exosomes for noninvasive diagnosis [3]. C. Yang et al. described a HEMA hydrogel-based, intelligent, wireless, theranostic contact lens for the in situ electrical sensing of intraocular pressure and the regulation of intraocular pressure via on-demand anti-glaucoma drug delivery [4]. 

### 3.2. Antibacterial CLs

CLs were reported as the leading cause of bacterial keratitis that results in visual loss. If they are not cleaned often enough, frequently, there are biofilms found on the CLs that lead to infectious corneal keratitis. Apart from improving the comfort of the CL wearer, efforts have been made to develop CLs with antibacterial abilities. Currently, there are several different strategies used to design CLs to kill bacteria and thus prevent corneal keratitis. These strategies are distinctive from each other in mechanism, yet they can be divided into two main methods. Figure 2 shows the strategies used to make contact lenses kill bacteria and thus prevent corneal keratitis. One is employing antibacterial agents in the CLs, which interfere with vital cell processes and damage the cell, leading to cell death. The other strategy prevents bacterial adherence on the surfaces of CLs, instead of killing bacteria directly.

#### 3.2.1. Applying Antibacterial Agents in CLs

To avoid the abuse of antibiotics and prevent the development of multi-drug-resistant bacteria, adding nanoparticles or peptides with antibacterial ability is a good option when fabricating CLs. Figure 3 shows examples of embedding nanoparticles or antibacterial peptides to modify CLs to kill bacteria. Nanomaterials such as gold, silver, and zinc oxide nanoparticles are popular materials as contact lens additives to protect against microbe infections. Apart from loading nanomaterials or molecules into contact lens gels, generating free radicals with a controlled and sustained rate is another favorable solution for bacterial keratitis treatment. 

##### Embedded Nanomaterials in CLs

The use of metallic nanoparticles based on their oxide features is of great interest. It is well acknowledged that their antibacterial activities are attributed to a few factors: (1) inhibiting enzyme activity; (2) triggering the production of ROS via the Fenton reaction, thus causing damage to cellular membranes, important proteins, lipids, and DNA. It is worth noting that certain metallic nanoparticles exhibit direct genotoxic activity [28]. In this review, we summarize the commonly used nanoparticles that are used as CL additives with antibacterial abilities, including silver nanoparticles, gold nanoparticles, and zinc oxide nanoparticles. 

Silver nanoparticles

Silver nanoparticles (Ag NPs) are recognized to possess a broad antibacterial spectrum, including Gram-positive and Gram-negative bacteria. It is generally acknowledged that the nanoscale size of Ag NPs makes them able to penetrate the bacterial cell wall and kill bacteria. However, the mechanism of Ag NPs in terms of antibacterial activity is not yet clarified. Currently, researchers agree that there are principally three mechanisms that work separately or combined to kill bacteria. The first mechanism proposes that due to their large ratio of surface area, they can increase the permeability of cell walls, inducing the leakage of cellular content and higher reactive oxygen production and stressing bacteria; they also interrupt deoxyribonucleic acid replication by releasing silver ions, which also contributes to bacterial killing [29,30]. The second claims that Ag NPs can continually release silver ions. Positively charged silver ions can interact with the negative cell membranes and cytoplasmic membranes, thus enhancing the permeability of cytoplasmic membranes and leading to the disruption of bacteria [30]. The last mechanism suggests that Ag NPs can interrupt important cell processes. Ag NPs have an ability to interact with sulfur or phosphorus groups, as sulfur and phosphorus groups are important components of deoxyribonucleic acid (DNA), and they can terminate DNA replication and cell reproduction [29,30]. Silver nitrate is commonly used as a prophylaxis tool for the intervention of neonatal ocular infection [31]. In addition, Ag NPs have been embedded in CLs. Some studies have proven their antibacterial efficacy regarding the solid inhibition of the biofilm formation of *P. aeruginosa* and *S. epidermidis* over 24 h of treatment, and they achieved more than 95% inhibition [32,33,34]. However, different authors claimed that Ag NP-embedded hydrogel CLs did not show significant differences in both *P. aeruginosa* and *S. epidermidis* strains [32,35]. In a randomized, double-blind pilot study of 60 subjects, Lakkis et al. reported that wearing Ag NP-infused CLs does not influence the microbe environment around the cornea [36]. A recent study showed promising results in the treatment of fungal keratitis both in vivo and in vitro with hydrogel CLs integrated with Ag NPs, quaternate chitosan, graphene oxide, and Voriconazole (an antifungal drug) [37]. Therefore, the use of Ag NPs as CL additives for corneal keratitis therapy remains for further investigation. Moreover, there is the potential toxicity of using Ag NPs for ocular applications [38,39]. 

Gold nanoparticles

Gold nanoparticles (Au NPs) have drawn significant interest in biomedical applications due to their unique properties, including tunable sizes, desirable functional groups for surface modification, and surface plasmon resonance. Thanks to their unique adsorption in the region of green light, Au NPs are used in CLs for red–green blindness management [40]. Au NPs have been demonstrated to have a broad antibacterial spectrum. Au NPs show different antibacterial effects compared to Ag NPs, as their shape, size, surface modification, and structure differ [41]. Due to their high affinity with cellular components and their specific surface area, they can attach to the bacterial membrane surface, affecting the stability and integrity of bacteria. Once Au NPs enter the bacterial cell, they can inhibit cell replication by interfering with DNA transcription and generating a large amount of ROS [41] to exert antibacterial activity. In addition, others describe different aspects of the antibacterial mechanism of Au NPs that include effects on apoptosis, electron transfer chain damage, and the disruption of metabolic pathways, e.g., the ATP production pathway [42]. Q. Guo et al. fabricated CLs with Au NPs to manage the kinetic release of ketotifen to treat conjunctivitis [43]. With the Au NPs inside the CLs, they achieved the higher uptake and reduced burst release of ketotifen. In addition, protein adherence on the Au NPs CLs was decreased. Similarly, F. A. Maulvi et al. loaded Au NPs into CLs and observed significantly higher timolol uptake. The in vivo study showed a significant improvement in drug deposition with the Au NP-embedded CLs in the ciliary muscle and conjunctiva [44]. Au NPs are generally recognized as nontoxic and have attracted interest in translational applications, such as cancer therapy and ophthalmology [45,46]. However, recent studies have shown that the toxicity of Au NPs is related to their size [47]. Thus, researchers should carefully consider their size when using Au NPs for biomedical applications. 

Zinc oxide nanoparticles

Zinc oxide nanoparticles (ZnO NPs) were reported to have a broad antibacterial spectrum and outstanding UV light adsorption ability. Zinc is one of the most well-studied biological objects; it has a very strong reduction ability to generate zinc oxide. Zinc is a very important trace element in the human body and plays vital roles; for example, it is distributed in human tissue and exhibits the highest concentration in myocytes [28]. The antibacterial effect of ZnO NPs shares some similar mechanisms with that of Ag NPs and Au NPs, consisting of the disruption of the cell membrane, the generation of ROS, binding to proteins and DNA, and the disturbance of DNA replication. ZnO NPs were reported to kill bacteria by altering a large range of genes’ expression, usually via the downregulation of gene expression [28]. Furthermore, the bacterial inhibition of ZnO NPs is impacted by their size, shape, concentration, and operating conditions. Different shapes of nanoparticles exhibit different surface areas, as nanoparticles interact with bacterial membranes depending on the effective surface area. ZnO NPs in spherical shapes were reported to release more Zn^2+^ compared to rod-shaped particles. The antibacterial activity of ZnO NPs occurs in a size-dependent manner: the smaller the particles are, the greater the depth at which they can penetrate inside bacteria. Size is considered the main reason for bacterial inhibition by ZnO NPs. The operating conditions, such as an acidic pH and high temperatures, facilitate the bacterial inhibition activity of ZnO NPs. High temperatures promote the aqueous solubility of ZnO NPs, and an acidic environment increases the release of effective Zn^2+^ irons [48]. The nanostructured oxides of zinc were demonstrated to have very strong antibacterial effectiveness at low concentrations ranging from 0.16 to 5.00 nmol/L against several bacterial strains [28,49]. U. Kadiyala et al. investigated the antibacterial activity of ZnO NPs against methicillin-resistant *S. aureus* (MRSA); in their study, in the presence of ZnO NPs, the growth of MRSA was dramatically decreased by an increase in ROS production and lipid peroxidation, and the significant upregulation of pyrimidine biosynthesis and carbohydrate degradation. At the same time, the amino acid synthesis in *S. aureus* was significantly downregulated, suggesting a complex mechanism of antibacterial activity [50]. ZnO NPs kill bacteria by inducing the generation of reactive oxygen species (ROS), which damage cell membranes and cause the leakage of intracellular DNA and proteins [51]. A. E. Nel et al. reported multifunctional CLs coated with ZnO, chitosan (CS), and gallic acid (GA). CS and GA were used to improve the comfort of CLs, with antioxidant properties and high wettability [52]. Combined with ZnO NPs, the CLs showed high antibacterial efficiency against *Staphylococcus aureus* (*S. aureus*) bacteria. A. K. Shakeel et al. observed similar significant bacterial and fungal inhibition results when functionalizing CLs with ZnO NPs [53]. A. Sung et al. proved that embedded ZnO NPs in HEMA CLs possessed excellent optical and UV-light-blocking properties [54]. Z. Zhu et al. [40] prepared ZnO/cyclized polyacrylonitrile CLs with an exceptional ability to block UV and blue light. In vivo experiments on ZnO/cyclized polyacrylonitrile showed strong bacteria-illing ability against Gram-positive (*S. aureus*) and Gram-negative (*Escherichia coli* (*E. coli*)) bacteria. The ZnO/cyclized polyacrylonitrile material was further proven to be nontoxic toward several cell lines, including human corneal epithelial cells, human umbilical vein endothelial cells, and L929 mouse fibroblast cells. Nevertheless, ZnO NPs can easily pass through the cell membrane and interact with cellular macromolecules to achieve therapeutic effects, and they were also found to induce oxidative stress and cause cytotoxic effects in some organs [28,55]. 

##### Integration of Antibacterial Peptides with CLs

Antibacterial peptides (AMPs) are a class of bioactive small molecules (<10 kDa) composed of less than 50 amino acid residues, which can be generated from the host’s defense system—the innate immune system that operates during infection procedures [57]. Their diverse biological activities have gained ongoing interest, especially regarding their bacteria-killing ability. The bacterial inhibition activities of AMPs are dependent on their shape, size, net positive charges, and amphipathic structures. These features of AMPs enable them to interact with bacterial surfaces and insert into lipid bilayers, leading to membrane rupture [57]. Due to their positive charges and amphipathic nature, antibacterial peptides contain a broad spectrum of antimicrobial abilities against various bacteria, viruses, and fungi. The positively charged AMPs allow them to interact with negatively charged bacterial membranes [58]. Furthermore, AMPs can also regulate immune systems to defend against escaped pathogens in the first line of defense [59]. Once attached to the bacterial surface, the amphipathic property of AMPs will enable them to enter bacterial cells and further rupture the bacteria [60]. There are a few approaches available to integrate AMPs into CLs. For instance, there are studies using EDC (1-ethyl-3-[3-dimethylaminopropyl] carbodiimide hydrochloride) coupling to covalently bond AMPs to CLs [59]. D. Dutta et al. covalently attached four different AMPs to HEMA hydrogel CLs [61]. Their results showed that Melimine and Mel-4 strongly inhibited the growth of *P. aeruginosa* and *S. aureus*. However, LL-37 only inhibited the growth of *P. aeruginosa*, and lactoferricin did not show any antibacterial activity. Similarly, E. Salvagni et al. functionalized CLs with two different short AMPs [62]. Bacterial studies demonstrated that AMP-functionalized CLs drastically reduced bacterial adhesion and viability against *P. aeruginosa* and *S. aureus*. The Mel-4 antibacterial peptide was added to silicone hydrogel CLs, and the Mel-4-coated CLs were found to display high antibacterial inhibition (>2 logs) ability [59]. There are studies that report that AMPs can directly interact with bacterial DNA and/or RNA, and thus affect their protein synthesis, replication, and translation processes. Melimine or Mel-4 presents very high antibacterial activity against *P. aeruginosa*. Melimine or Mel-4 was also found to cause 75% and 36% cellular ATP release after 2 min of interaction. After 5 min of interaction, bacterial membranes were damaged, together with simultaneous DNA/RNA release [56]. AMPs were also found to be able to inhibit viral spread by directly interacting with membranous viral envelopes and molecules on the host cell surface [63]. Overall, AMPs are proven to be an excellent therapeutic candidate for infectious diseases. 

##### ROS Produce Reagents for Bacterial Killing 

ROS play a crucial role in killing bacteria. They are highly reactive and extremely unstable; they can damage bacterial membranes, lipids, and DNA and eventually damage bacterial cells. ROS targeting is a promising treatment alternative compared to drugs or antibiotics. For example, organo-selenium (Se) has been used to generate superoxide radicals (O_2_*) and hydroxyl radicals (OH*) to destroy bacterial cell membranes, organelles (e.g., mitochondria, lysosomes, etc.), DNA, proteins, and lipids and damage bacteria. Thus, embedded agents in CL hydrogels can generate ROS under certain circumstances, which is an alternative approach for corneal keratitis. Organo-selenium is an important trace micronutrient for almost all living creatures. For example, selenium plays an essential role in regulating the metabolism of thyroid hormones [64], and low selenium levels may be related to thyroid disease. In recent years, selenium has drawn attention in biomedical applications, especially in utilizing its antibacterial properties for manufacturing medical devices. Selenium nanoparticles can be prepared by different methods, with high biological activity and photoelectric performance [65]; they have been used in various applications, such as wound bandages, catheters, dental sealants, and contact lenses, to inhibit bacterial biofilm formation and therefore prevent infection [66]. The sealant that contains selenium was reported to eliminate the formation of biofilms from oral bacterial strains including *S. mutants*, *S. salivarius*, and *S. sanguinis* [66]. Selenium-coated silicone hydrogel CLs did not exhibit the formation of biofilms, and the antibacterial activity of selenium-coated CLs lasted for 2 months. Animal experiments on rabbits did not show any side effects after 2 months of usage [66]. Nitric oxide (NO*) is an important signaling molecule that controls physiological functions; its low molecular weight and fast diffusion pose a significant challenge for NO* delivery [67]. Although NO* therapy may offer a benefit for bacterial infectious keratitis, due to the poor stability of NONOates, uncontrolled NO* release can occur. Therefore, it is of great interest to deliver a controllable amount of NO* to the lesion site. J. Aveyard et al. described a hydrogel contact lens with the controlled release of NO* for more than 15 h [68]. Figure 4 below shows the functionalized CLs with sustained NO* release and demonstrated a significant antibacterial capacity against *P. aeruginosa* and *S. aureus*. The in vitro cell experiment with corneal epithelial cells proved that the contact lens gels were nontoxic, suggesting a viable option for corneal keratitis. Although using ROS as a tool to combat bacteria has been demonstrated to be useful and efficient, the concentration of ROS must be carefully managed, as high ROS might cause topical oxidative stress to human tissue. It is worth mentioning that free radicals, such as OH*, O_2_*, or NO*, have extremely short lifetimes; only when bacteria are localized very close to the free-radical-releasing agent can they be damaged. 

#### 3.2.2. Prevention of Bacterial Adherence to CLs

Planktonic bacteria tend to attach to the surfaces of CLs and form biofilms. Once the biofilms are established, they are challenging to eliminate. Instead of using active methods to kill bacteria, an alternative method to avoid bacterial infection is to modify the CLs’ surfaces to prevent bacteria from attaching to CLs. Bacteria adhere to the surfaces of CLs by producing an adhesion factor. The bacterial strains and CL surfaces were studied regarding their chemical and physical properties to contribute to the understanding of bacterial adhesion on CLs. The hydrophobicity of CL hydrogels can support certain bacteria strains [69]. Hydrophilic hydrogel-based CLs tend to adapt more to hydrophilic bacteria, whereas CLs with hydrophobic substrates are suitable for hydrophobic bacteria strains [69]. The surface roughness of CLs was reported to be not only related to comfort but also bacterial adhesion [70,71]. However, convincing data are required to support this argument. L. Kodjikian et al. [72] investigated the bacterial adhesion on standard HEMA and silicone hydrogel CLs, finding remarkably fewer bacteria attached to standard HEMA CLs. Their results suggest that HEMA gel is a suitable material as a CL to reduce bacterial attachment. A. M. Rediske et al. [73] reported that soft hydrophilic CLs in the presence of human polyclonal immunoglobulin (IgG) could significantly reduce the adhesion of *P. aeruginosa* at a concentration of 25 mg/mL. Meanwhile, diluted IgG at the concentration of 10 mg/mL did not affect bacterial adhesion after 2 or 4 h. Comparably, A. Mordmuang et al. claimed that HEMA hydrogel-based CLs are suitable for bacterial attachment and biofilm formation; in their study, *S. aureus*, *P. aeruginosa*, *E. coli*, and *K. pneumonia*—four bacterial strains—were tested, and *S. aureus* and *P. aeruginosa* were found to display the highest biofilm formation due to bacteria on HEMA-based CLs [70]. 

## 4. Conclusions

Bacterial infections have always been the most critical public health concern, as they induce life-threatening and severe diseases, such as infectious corneal keratitis. The eyes are vital sensory human organs, and infectious corneal keratitis can result in blindness within days without immediate and appropriate treatment. Though many contagious bacterial diseases are targeted with the development of antibiotics and public hygiene, the global overuse of antibiotics has led to the emergence of multi-drug-resistant bacterial strains. Therefore, there is an urgent need to develop novel approaches to fight against bacterial resistance and treat bacterial infections with high effeteness. CLs play an essential role in our daily lives and fulfill many therapeutic purposes. Thus far, CLs have demonstrated their use as medical devices in various ways—for instance, by killing bacteria and preventing corneal keratitis. However, a more comprehensive evaluation of polymerized CLs would be useful before they are applied in clinical trials. CLs tend to adsorb proteins (e.g., lysozyme, albumin, and IgG) from the tear liquid; when these proteins accumulate on the surfaces of CLs, they are inclined to cause foggy vision. The eyes may recognize some materials of the CLs as foreign materials and trigger inflammation, thus stressing the cornea. Subsequently, an excessive oxidative stress burst will occur, and many reactive oxygen species will be generated and damage the cornea.

## Figures and Tables

**Figure 1 pharmaceutics-14-02635-f001:**
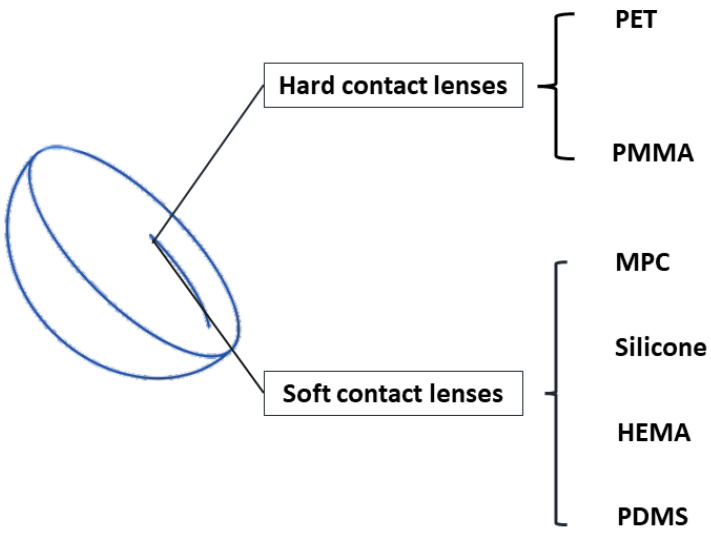
Hydrogel types for CLs.

**Figure 2 pharmaceutics-14-02635-f002:**
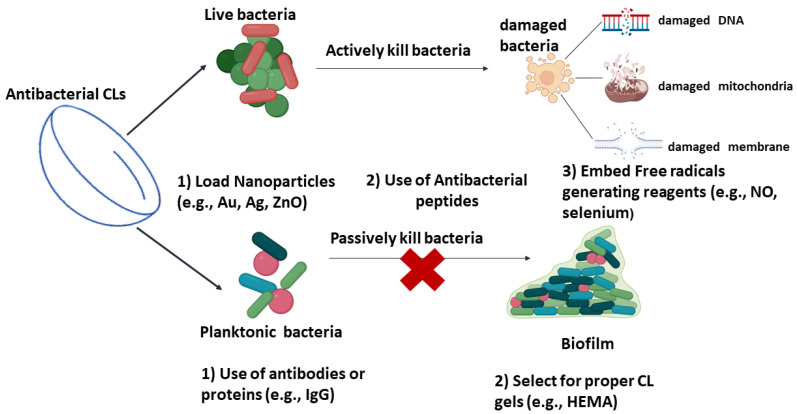
Strategies applied in CLs for bacterial keratitis.

**Figure 3 pharmaceutics-14-02635-f003:**
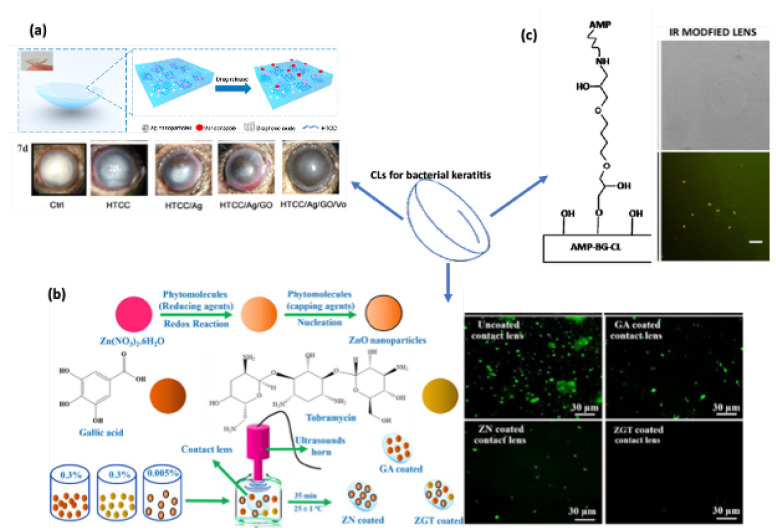
Examples of functional CLs for corneal keratitis. (**a**) Ag NP-embedded CLs. Reproduced with permission of reference [38]. (**b**). ZnO NP-loaded CLs. Reproduced with permission of reference [54]. Scale bars: 30 um. (**c**) AMP-coated CLs Reproduced with permission of reference [56]. Scale bars: 10 um.

**Figure 4 pharmaceutics-14-02635-f004:**
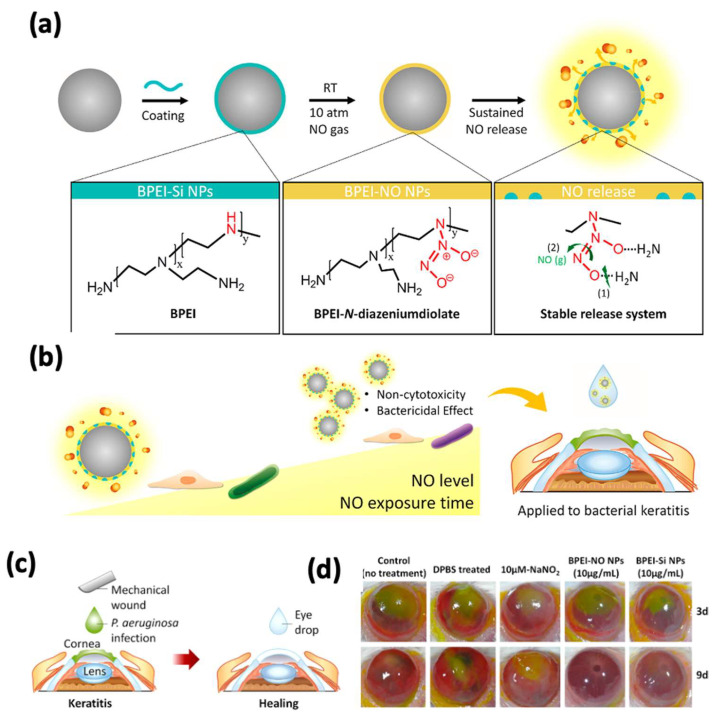
Antibacterial contact lenses with NO* generation. (**a**) Schematic illustration of functionalized CLs. (**b**) Illustration of the effects of CLs on cells and bacteria depending on the concentration and exposure time of NO*. (**c**) Schematic illustration of wound healing experiment. (**d**) Images of mouse corneas treated with CLs. Reproduced with permission of reference [68].

**Table 1 pharmaceutics-14-02635-t001:** Molecular formula and properties of commonly used hydrogels for CLs.

Material		Molecular Formula	Advantages	Disadvantages	Reference(s)
Hard CLs	PMMA	(C_5_H_8_O_2_)_n_	Excellent optical properties, high toughness, rigid, inexpensive	Low gas permeability, limited hydrophilic ability, inflexible	[16,17]
	PET	(C_10_H_8_O_4_)_n_	Outstanding chemical and thermal resistance, inexpensive	Low hydrophobic ability, glass transition temperature, rigidity, and surface energy	[16]
Soft CLs	MPC	C_11_H_22_NO_6_P	Good surface wettability, high gas permeability, low protein adsorption, and inexpensive	Low mechanical strength	[17]
	PDMS	(C_2_H_6_OSi)_n_	High gas permeability, inexpensive	Limited wearability	[16,18]
	HEMA	(C_6_H_10_O_3_)_n_	High water content, good chemical and thermal stability, high hydrophobic ability, flexibility, gas permeability, biocompatibility, and inexpensive	Low gas permeability, protein deposition issues	[16]
	Silicone	(R_2_SiO)_n_	High gas permeability	Highly hydrophobic, expensive	[19]

## Data Availability

All data supporting reported results can be found in this manuscript.

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
