# Peer review of "Recent Applications of Contact Lenses for Bacterial Corneal Keratitis Therapeutics: A Review"

_pharmaceutics, 2022, doi:10.3390/pharmaceutics14122635_

Round 1

Reviewer 1 Report

The manuscript is about functional contact lenses for antimicrobial applications for eye infections. Bacterial keratitis was defined, and its diagnosis and traditional therapies were explained. Controlled drug release system integrated in the contact lenses were discussed for hydrogels. Figure 1 is a schematic for commonly used hydrogel components applied in contact lense applications and their main properties in Table. Strategies to prepare antimicrobial contact lenses were explained around the methods used: embedded nanomaterials, antimicrobial peptides, ROS producing reagents, preventing adherence of bacteria. I recommend that the manuscript needs visual aids in explaining antimicrobial contact lenses, like figures from published examples.

Author Response

Response: The reviewer has a good point. We did include a figure from published articles in Figure 3. The figure explains antibacterial contact lenses embedded with nanoparticles like silver nanoparticles, antibacterial peptides, and Zinc oxide nanoparticles separately. We also added a figure (Figure 4) illustrating contact lenses that kill bacteria via ROS generation.

Reviewer 2 Report

Good review summarising current diagnosis, traditional treatment and recent application of contact lenses-based antibacterial application. Definitely can serve for any readers in orientation in this developping issue. I have no remarks.

Author Response

I would like to thank this reviewer for the valuable suggestion